# Scoping Review Protocol of Technological Interventions for Vocational Inclusion of Individuals with Disabilities

**Sara Hamideh Kerdar [1],\*, Britta Marleen Kirchhoff [1], Liane Bächler [2] and Lars Adolph [1]**

1 Federal Institute of Occupational Safety and Health, 44149 Dortmund, Germany
2 Department of Rehabilitation and Special Education, University of Cologne, 50674 Cologne, Germany
\* Correspondence: hamidehkerdar.sara@baua.bund.de

**Abstract:** Technology could improve the vocational inclusion of people with disabilities, thus increasing their wellbeing and competence development. Moreover, societies could benefit from their skills and expertise. In this protocol, the objectives, structure, and further details of a scoping review on the subject of the vocational inclusion of people with disabilities via technologies are described. This article additionally demonstrates how a piloting phase can be used for the further development of the protocol. The focus of the proposed scoping review is disability, technology, and task/work. Expansive and specific keywords will be searched in APA PsycInfo, APA PsycArticles, and CINAHL Complete via EBSCOhost, Web of Science, Embase, Scopus, and IEEE Xplore. As regards the grey literature, ProQuest will be used for dissertations and theses and Google Scholar will be hand searched. Articles published in 2012–2022 focused on working-age adults will be exported to EndNote and titles/abstracts will be monitored. We further describe the inclusion and exclusion criteria, data extraction, and charting strategies of the proposed scoping review. The results will be mapped and reported based on disability, technology, and task. For the improvement of the protocol, a pilot study in February 2022 was performed. The results from the pilot, briefly reported herein, led to a transparent and clear structure of the proposed scoping review.

**Keywords:** disability; technology; work; inclusion; employment; occupational inclusion; digital technology; assistive technology

## 1. Introduction

Work, necessary to making a living in the current economic structure, can enhance individual wellbeing [1]. The employment of people with disabilities both provides them with the chance of being economically independent [2] and increases their quality of life [3–5]. Well-designed work can also improve mental health and reduce the chance of stigmatization in work environments [6]. Even though the employment rate of people with disabilities varies in different countries, they are underemployed globally and have fewer opportunities for employment in comparison with people without any disabilities [7,8].

### 1.1. Definition of Disability and Inclusion

It is challenging to find a universal definition and classification of disability, as a definition concerns various factors, including cultural [9] aspects. Centers for Disease Control and Prevention (CDC) defines disability as "any condition of the body or mind (impairment) that makes it more difficult for the person with the condition to do certain activities (activity limitation) and interact with the world around them (participation restrictions)" [10]. This definition summarizes the model presented by the International Classification of Functioning, Disability and Health (ICF) of the World Health Organization (WHO) [11], in which environmental factors, including technology and products, are specifically taken into consideration. Similarly, the United Nations' (UN) Convention on the Rights of Persons with Disabilities (CRPD) defines disability as the "results from

the interaction between persons with impairments and attitudinal and environmental barriers" [12]. Such approaches incline to the social model of inclusion, wherein individuals face disabilities due to a society's physical barriers, stigma, lack of rights, technologies, and resources [13]. Lollar and Crews argue that the evaluation of environmental factors is needed in public health to increase the inclusion of people with disabilities in society [14]. Movements such as the UN's convention have initiated regulations and laws for providing occupational opportunities for people with disabilities internationally. On the subject of work and employment, the UN emphasizes that "the right of persons with disabilities to work, on an equal basis with others" should be recognized. These individuals should have "the right to the opportunity to gain a living by work freely chosen or accepted in a labour market and work environment that is open, inclusive, and accessible to persons with disabilities" (UN 2006, Art. 27, para. 1) [15].

### 1.2. Technology and Work

Technology advancements have changed the nature of work and, despite their potential disadvantages (e.g., work overload or stress) [16], they could facilitate the inclusion of people with disabilities in work environments. For instance, assistive technologies (AT) could support people with intellectual disabilities (ID) in the workplace [5]. Morash-Macneil et al. argue that wearable technologies such as smart watches or portable electronic devices such as iPhones could increase the independence of people with ID at work [5]. Technologies such as "refreshable braille displays" or "screen readers" provide chances for people with visual disabilities to use text-editing technologies for work or studies [17]. Tomczak believes that AT could provide appropriate solutions for the obstacles individuals with autism spectrum disorder (ASD) face in the workplace. For example, by using emails or online platforms, problems with interpersonal communications could be prevented (as direct and verbal communication could be replaced through these platforms) [18]. Kildal et al. designed and piloted a collaborative robot for assembly workers with cognitive disabilities [19]. Through robot–person interactions, they reported promising results in terms of the integration of people with cognitive disabilities in assembly work positions. Another example of using technology to support people with disabilities is the integration of sign language in online video software. Janeera et al. formulated an algorithm in which the hand gestures of deaf and mute people are interpreted for all the participants of the video call [20].

### 1.3. Disability, Work, and Technology

In an overview of the literature, Jurado-Caraballo et al. highlighted the fact that "disability and work" is a far less common focus as compared with other perspectives such as gender or race [21]. This gap is also observed regarding technology and work. Many studies focus on technology and disability (e.g., the use of technology in rehabilitation [22], education [23–25], and diagnoses [26,27]); however, the role of technology in the occupational participation of people with disabilities remains scarce. This is in accordance with the literature review by Lian and Sunar, which showed that studies on mobile augmented reality technologies for ASD were mostly conducted in schools or labs and were mostly focused on social activities, education, behaviour enhancement, and preparations for job interviews [28]. Moreover, in a systematic review, Shattuck et al. found that most of the articles were related to employment and the impact of AT on work for people with ASD [29]. It is noteworthy that the majority of reviews either concentrate on a specific type of disability (e.g., ASD) or on a specific type of technology (e.g., AT). Therefore, the current state of the literature regarding technology, disability, and work from a wider perspective is unclear. In a report on disability, exclusion, and inclusion, Rohwerder reported that environmental factors and the lack of knowledge about what exactly works, among other things, could be barriers to the inclusion of people with disabilities [30].

Scoping reviews are an appropriate means of identifying, mapping, and reporting the existing knowledge and gaps within the literature [31]. Thus, by conducting a scoping

review regarding the vocational inclusion of people with disabilities via technologies, we intend to review the literature and identify what types of disability, technologies, and work activities have been studied over the years. Through the proposed review, technologies that could increase the chances of the occupational participation of people with disabilities will be highlighted. Additionally, the outcomes and key elements of (successful) technological implementations in work environments alongside factors that could influence the vocational inclusion of people with disabilities will be investigated. For instance, it is important to know what additional elements, such as office environment, pre-training of co-workers, etc., could be contributing factors to successful vocational inclusion. For better quality, transparency, and trustworthiness of the proposed review [32], the priori protocol is explained in detail. This paper additionally demonstrates how a piloting phase can be used for the further development of the protocol.

## 2. Materials and Methods

The proposed scoping review is part of a project of the Federal Institute for Occupational Safety and Health in Germany. Its prospective results should represent a step towards the humane design of work for people with disabilities with the help of innovative technologies. The project and the scoping review are available on the Open Science Framework (OSF) (https://osf.io/quwd5/ (accessed on 19 August 2022)).

The proposed scoping review will be carried out in accordance with the PRISMA extension for scoping reviews (PRISMA-ScR) [33]. Additionally, the review will follow the framework introduced by Arksey and O'Malley for scoping reviews [34]. Therefore, the following five stages will take place:

1. Identifying the research question;
2. Identifying relevant studies;
3. Study selection;
4. Charting results;
5. Collating, summarizing, and reporting results.

Moreover, the improvements by Levac et al. [35] and Joanna Briggs Institute's (JBI) framework enhancements [36] will be taken into consideration. Therefore, as suggested by JBI [31], an initial draft of a protocol (i.e., a plan) for the proposed scoping review was developed. In order to establish the consistency of the steps and further preparations, a pilot study was conducted [31]. The piloting process led to the enhancement of the protocol and methods, namely, it provided a clear definition of the inclusion and exclusion criteria, keywords, and the objectives of the proposed scoping review. In the coming sections of this paper, we explain the steps and structure of the proposed scoping review in detail. Thereafter, the methods and results of the pilot study will be presented.

### 2.1. Stage 1: Identifying the Review Question

The core question of the proposed review will focus on "how technologies could facilitate vocational inclusion of people with disabilities?" Additionally, as technology, disability, and work are the main variables, the review questions are as follows:

1. What technological interventions for occupational inclusion of people with disabilities have been studied?
2. What types of disabilities have been addressed for the occupational inclusion of people with disabilities using technological interventions in the workplace or work-related activities?
3. What type of activity or field of work has been the focus of studies concerning the impact of technology in the workplace for the occupational inclusion of people with disabilities?
4. What additional factors could influence the vocational inclusion of people with disabilities when technologies were implemented?

*2.2. Stage 2: Identifying Relevant Studies*

Articles in English and German will be included. The following databases are intended to be searched: APA PsycInfo, APA PsycArticles, and CINAHL Complete via EBSCOhost, Web of Science, Embase, Scopus, and IEEE Xplore. However, based on the initial search results, additional databases could be also considered. As regards grey literature, data sources such as ProQuest for dissertations and theses and Google Scholar will be hand searched. A wide range of keywords will be applied for the search, in order to expand the search results. An example of the keyword search is presented in Supplementary File S1. Articles will be imported to EndNote X9 and their titles and abstracts will be monitored.

Inclusion/Exclusion Criteria

Technology: Technologies could be categorized into low (e.g., tape recorders), moderate (e.g., digital watches), and high technology (e.g., tablets and computers) [37,38]. For this review, we will consider studies that have used moderate and high technologies only, for example, assistive, computer-based technologies, robotics, etc. Additionally, based on the fact that technology advances rapidly, the proposed review will be limited to the last 10 years (i.e., 2012 to 2022). We will also only consider studies on working-age participants (16 to 67 years old). Thus, no children or students or studies preparing young adults for job interviews will be included.

Disability: The proposed scoping review will focus on developmental, physical, and multiple disabilities.

Work: The word "work" denotes several types of employment [39], including paid or unpaid work, self-employed or hired work, etc. In the proposed scoping review, all types of work and employment will be considered, as the purpose of this study is to find technologies that facilitate the occupational inclusion of people with disabilities. Additionally, tasks such as assembly work or food preparation in work setups will be included.

Study selections: All types of study designs will be taken into consideration. In particular, specifically adapted technologies such as collaborative robots are sometimes only investigated in individual case studies, from which relevant insights can nevertheless be derived. Despite reviews being excluded, their literature will be hand searched and relevant studies, if any, will be included. The majority of conference abstracts have incomplete information and the authors may not publish any articles pertaining to these abstracts; however, they remain valuable sources of information. Therefore, conference abstracts will be examined. If they fail to provide enough information for the review, the authors will be contacted to request full-text articles on the same topic. If the full texts are available or the abstracts contain sufficient information, then they will be included.

*2.3. Stage 3: Study Selection*

As suggested by Levac et al., discussion about inclusion and exclusion of articles in the beginning stages of the scoping review is very important [35]. Thus, once the search data are exported, two independent reviewers will monitor a sample of 200 randomly chosen articles. The inter-rater reliability will be calculated using Cohens Kappa for the degree of agreement. Upon satisfactory results, one reviewer will carry out the review process and uncertainties will be discussed within the team throughout the whole process [30]. Once all articles are monitored, abstracts from the included articles will be monitored by two reviewers and once a common decision is made, the full text of included articles will be studied and the charting process will begin.

*2.4. Stage 4: Charting the Data*

In an Excel table, the following subjects for the included articles will be recorded: author(s), year of publication, country of origin, study design and setting, sample size, disability, (type of) technology, task, results, and other influential factors (e.g., pre-training of co-workers, presence of a job coach, personality traits, etc.).

*2.5. Stage 5: Collating, Summarizing, and Reporting Results*

The included studies will be grouped and reported based on (a) disability, (b) type of technology, and (c) type of task. However, based on the articles included, changes and enhancements might be necessary [31]. For instance, as the categorization of technologies usually varies, depending on the results, new categorizations of technologies might be introduced.

**3. Pilot Methods and Results**

In February 2022, the keywords "technology AND work OR workplace OR employment AND disability" were searched in Embase. Studies published in 2010–2022 focused solely on adults were included (i.e., studies conducted on children or students were excluded). The search results ($N = 7476$) were imported to EndNote X9 and duplicate articles ($N = 212$) were deleted. Titles and abstracts were monitored. To enhance the inclusion criteria, two reviewers monitored the title and abstracts of articles that created doubt until a common decision was made. After monitoring 7476 articles, 15 studies matched the inclusion criteria. Once a final decision was made among authors concerning the charting details, the full texts of the included articles were monitored and the data were charted. The charting table is available in Supplementary File S2.

It is noteworthy that, after reading the full texts of the articles, we excluded three articles. The excluded articles and the reasons of exclusion are as follows: Gunther et al. designed and prototyped a high-frequency radio frequency identification (UHF RFID) system for employees with cognitive disabilities who work in warehouses [40]. Despite the fact that their study was designed for people with disabilities, the prototype was conducted on participants without a disability; hence, the study was excluded. We also excluded the survey conducted by Lin et al., in which 132 people with disabilities answered a questionnaire [41]. This survey demonstrated that the home-based employment services enhanced the employment status of people with disabilities. This study also showed that online working (such as e-commerce or internet marketing) increased the chances of inclusion. However, as the study did not clarify the means (i.e., the technology) used by people with disabilities, we excluded this survey from our results. Lastly, in a conference abstract, Miyazaki et al. presented the advantages of smart glasses (using artificial intelligence) for people with ASD in the workplace [42]; however, we excluded this study as the abstract did not provide sufficient details regarding the study. A search for an article relating to the abstract was unfortunately unfruitful, as was contacting the authors.

As the focuses of both the pilot study and the proposed full scoping review are the three variables of disability, technology, and task, the results of the pilot study, as a trial, are presented accordingly (Table 1). It is noteworthy that the piloting process took place to enhance this protocol, in order to finalize the best approach before conducting the full review. Hence, no conclusions should be made based on the pilot's results.

**Table 1.** Results of the pilot study, based on disability, technology, study setting, and task.

| Disability Group | Author | Disability | Technology | Study Setting (S) and Task (T) |
|---|---|---|---|---|
| **Developmental disabilities (DD)** | Ertas et al. (2020) [43] | Intellectual disability | An action planning app as an assistant | S: Home-office<br>T: (unspecified) 2 tasks every day |
| | Allen et al. (2012) [44] | Autism and intellectual Disabilities (ID) | Video modelling training, audio cuing using headphones paired with cell phones | S: Retail store<br>T: Promoting products by wearing an air-inflated WalkA-round® costume of a popular commercial character |
| | Bross et al. (2019) [45] | Autism Spectrum Disorder (ASD) | Video modelling (videos were recorded using an iPhone and were shown using a laptop) | S: Retail store<br>T: Cashier |
| | Bross et al. (2020) [46] | ASD | Video modelling (videos were viewed on a laptop) | S: In a large grocery store<br>T: Grocery store courtesy clerk |
| | Chezan et al. (2020) [47] | ASD and co-occurring moderate ID | Audio coaching using Smartphone, Bluetooth headset, earbud speaker headset with built-in microphone | S: Replication of an office in a university classroom<br>T: Conversation and self-initiated interactions toward coworkers |
| **Physical disabilities** | Ferronato and Ukovic (2014) [48] | Vision impairments | Assistive technology (AT) (software and talking products) | S: Participants' workplace (office and kitchen)<br>T: Working and accomplishing their duties at their workplace |
| | Pruettikomon and Louhapen-sang (2018) [49] | - Physical disability<br>- Visual impairment<br>- Hearing impairment | Office adjustments (such as hydraulic adjustments for tables and cabinet compartments for wheelchair), and an app to assign tasks to disabled staff) | S: Retail and wholesale companies<br>T: Working and accomplishing their duties in their workplace |
| | Luquini et al. (2020) [50] | Inflammatory arthritis | Online self-management program (online self-learning modules and group meetings, individual vocational counselling, and ergonomic consultations) | S: Not specified<br>T: Not specified |

Table 1. *Cont.*

| Disability Group | Author | Disability | Technology | Study Setting (S) and Task (T) |
|---|---|---|---|---|
| **Multiple disabilities** | Lancioni et al. (2013) [51] | Blindness and ID | Technology system: audio, sound, sensors | S: A center for persons with multiple disability<br>T: Assembling trolley wheels |
| | Lancioni et al. (2014) [52] | Low vision or total blindness, severe/profound ID, and minimal object interaction | Computer system, interfaced with optic sensors and controlled the delivery of visual and auditory stimulations (song and videos) | S: Activity room of an education center<br>T: Constructive object-manipulation responses |
| | | Deafness, severe visual impairment, and profound ID | A box with strobe light and an optic sensor, a vibration device with light, a chair with back massage, a remote-control device radio linked to the boxes, optic sensors, and the stimulation devices | S: Activity room of a rehabilitation and care center<br>T: assemble a five-component water pipe |
| | Morse et al. (2021) [53] | - ID, visual, speech disability<br>- ASD, language and speech impairment<br>- ID, post-traumatic stress disorder, cerebral palsy, language impairment<br>- ASD and language impairment<br>- ID and language impairment<br>- ID, orthopedic, speech, and language disabilities | An app in an iPad (located on a stand where participants could watch videos and see pictures and instructions of making smoothies) and a blender to make the smoothies | S: Participants were recruited from an agency that provided supported employment and community living program for adults with developmental disability. The study was conducted in the agency's staff break room.<br>T: Smoothie preparation |
| | Chang et al. (2012) [54] | Cognitive disabilities:<br>- Traumatic Brain Injury (TBI), physical impairments<br>- Intellectual and developmental disability (IDD)<br>- Schizophrenia<br>- TBI, Dementia<br>- Organic brain syndrome, epilepsy, IDD | A model consisting of:<br>- Personal Digital Assistant (PDA; AT)<br>- Bluetooth beacon sources to trigger instructions to PDAs<br>- A prompting engine to create unique instruction per individuals based on the user profile, job schedules, and task lists | S: In a community-based coffee shop mainly operated by staff with cognitive disabilities<br>T: To complete 3 orders (pick up to 5 different items ordered, walk it to the table, walk to cashier window):<br>- Order of desserts<br>- Order of beverages<br>- Order of cookies |

## 4. Discussion

Due to the lack of clarity on the subject of the vocational inclusion of people with disabilities through technological interventions, a scoping review is a suitable method with which to understand the current state of the literature. Additionally, these reviews are the best way of summarizing "breadth of evidence" [35]. Moreover, through scoping reviews, the possibility of monitoring all types of studies, including the grey literature, is possible, hence the possibility of bridging knowledge [34,35]. As a result, through an extensive search, the proposed scoping review will identify the current perspective of the literature and the extent of the knowledge [31] on the subject, highlighting the gaps.

Protocols are a valuable addition to scoping reviews as they not only provide more clarity and transparency of the full review, but also prevent biases [32]. Additionally, piloting the protocol steps improves the details and facilitates better planning. We decided to present the pilot's results in this protocol, as they were a great means of enhancing the methods and clarifying the search strategy of the proposed review. The piloting phase highlighted that studies and consequently articles' titles and abstracts report specific types of disabilities and/or technologies. Hence, we developed extensive and detail-oriented keywords for the full review (an example is available in Supplementary File S1). We also learned that covering all types of disabilities for one single scoping review is impractical, as there are multiple types of disabilities caused by a variety of factors. Hence, we narrowed down and specified the inclusion and exclusion criteria in the piloting phase. Monitoring and mapping articles in the pilot process helped us to decide which types of articles should be included in the proposed review. For example, we learned that conference papers/abstracts could be valuable sources of information; however, there are few published in full-text formats. These insights helped us to decide how to deal with such situations and consequently provided a better, more detail-oriented protocol. Considering the fact that scoping reviews are still in need of enhancement [35], performing a pilot study in the protocol phase is beneficial.

One of the biggest challenges of the proposed review is the lack of a global classification of disabilities. The ICF argues that any individual could face some sort of disability during their lives; thus, the focus should be on a person's functionality and not the disability itself [11]. Even though such definitions could reduce discrimination and stigma, the lack of a definition of a disability hinders the chances of inclusion of people with disabilities [14]. We experienced these inconsistencies while summarizing the results of the pilot study. For example, the study of Chang et al. reported the disability of the study's participants as cognitive disabilities [54]. However, they recruited participants with ID (Table 1). Even though intellectual disabilities could be accompanied by cognitive deficits [55], they could be categorized separately. The Diagnostic and Statistical Manual of Mental Disorders (DSM), for example, categorizes ID under neurodevelopmental disorders and specifies a different category of neurocognitive disorders for cognitive disabilities [56]. This lack of a global classification could create confusion in research and thus the inability to replicate studies.

Another challenge faced in the piloting phase was the uneven description of technologies used in the studies. For instance, some articles (e.g., Morse et al. [53]) describe their research method and process in great detail. On the other hand, the included conference abstracts offer a short description of the technologies used. Another example is the abstract from Luquinit et al. in which they introduce an online self-management program. However, it is not clear whether all the services, such as group meetings and consultations, were also held online and which tools were used [50].

We anticipate several limitations of the proposed scoping review. Firstly, in order to include both dated and advanced technologies, articles published in the last decade will be included. This could potentially present limitations of excluding relevant studies conducted in previous years. Another limitation of this review is the language criteria, i.e., English and German, which rules out articles in other languages. Lastly, there are many technologies that have been developed and studied. Finding all the articles regarding all the technologies is not possible. However, we will try to expand our search by using extensive keywords.

## 5. Conclusions

People with disabilities should be considered as individuals with specialities and skills [9] who can provide additional value for both organizations and societies. Technologies have the potential to increase the inclusion of individuals with disabilities in the future. In this protocol, we provide a detailed plan for a proposed scoping review on the subject of the vocational inclusion of people with disabilities through technology. The aim of the subsequent scoping review is to identify and explore the available knowledge on the subject. The results of the review will highlight the gaps and map the way in which the literature is currently organized in order to finally suggest directions for future studies. As the scope of disability and technology is very wide, we performed a pilot study to enhance the search strategy, the inclusion and exclusion criteria, and the review's objectives. The piloting phase led to a transparent and detail-oriented protocol for the proposed scoping review.

**Supplementary Materials:** The following supporting information can be downloaded at: https://www.mdpi.com/article/10.3390/disabilities2030038/s1, File S1: Keyword example for the full review; File S2: Pilot study charting results. References [43–54] are cited in the supplementary materials.

**Author Contributions:** Conceptualization, B.M.K. and L.A.; methodology, S.H.K. and B.M.K.; formal analysis, S.H.K.; investigation, S.H.K.; resources, S.H.K.; data curation, S.H.K.; writing—original draft preparation, S.H.K.; writing—review and editing, S.H.K., B.M.K., L.A. and L.B.; supervision, B.M.K. and L.A. All authors have read and agreed to the published version of the manuscript.

**Funding:** This research received no external funding.

**Institutional Review Board Statement:** This scoping review/protocol focuses on previous studies; hence, an ethical approval is not required.

**Informed Consent Statement:** Not applicable.

**Data Availability Statement:** Not applicable.

**Conflicts of Interest:** The authors declare no conflict of interest.

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
