# Peer review of "Scoping Review Protocol of Technological Interventions for Vocational Inclusion of Individuals with Disabilities"

_disabilities, doi:10.3390/disabilities2030038_

Round 1

Reviewer 2 Report

Thank you to the authors for preparing an interesting manuscript on an important area.

This paper describes a scoping review protocol, and a pilot implementation of that protocol, concerning technologies for persons with disabilities and their intersections with work. 

The background explores the meaning of work, some definitions, and intersections between technology and work for people including persons with disabilities. 

Methods for both a pilot and a full study, yet to be completed, are presented. 

Results tabulate literature found during the scoping phase, and a discussion explores these results and presents some interpretation on the use of a scoping review itself. Challenges for a proposed full review are noted. The work concludes with a re-stating of the importance of disability inclusive work and the role of technology, but says nothing of the work conducted here, or the proposed review. 

This latter point illustrates my main concern with this manuscript. What has been done is valuable. What will be done is unclear, and how it builds on or learns from the work here isn't adequately described. The paper is neither a protocol nor a scoping review, or both. A fairly subtantial re-think is needed, but ultimately this is successful as a preliminary (if comprehensive) scoping review.

As a protocol, I was expecting a the initial scoping review to lead to a clearly stated search strategy and expanded framework for data extraction, or a similar way to describe what was learned from the scoping review about a full review. 

As a scoping review, I was expecting to read some integrative conclusions about the main question, which seems to be 'how might technology faciliate work for persons with disabilities?', as well as the scoping question of 'what does the literature tell us about this phenomenon, and how is literature organised', and finally 'how might a full review be structured to explore these themes more fully' 

I find that none of these questions are answered clearly, but that the work done is uniquely valuable and timely, and should be restructured. The structure moves from 'we did' to 'we will' throughout, which is hard to follow. 

If the authors find value in a pilot review to develop a protocol, that should be the focus. State a question to match. 'What should a protocol be to explore the role of technology in work for persons with disabilities' or similar. If the value here is making sense of diverse literature, then this is a standalone paper which could conclude quite reasonably that additional reviews are valid, and can benefit from both the content (what the literature says) and the structure (what variables are reported for what contexts) reported in this pilot review. 

As it is, the paper is unclear on both fronts, and a revision is required. 

I would strongly encourage that revision, noting the scope of work and the very unique contribution this review can make to understand how technology can support work in persons with disabilities, and how the literature describes this phenomenon. 

The alternative is a simpler, shorter protocol paper that spells out the proposed review, with a much shorter description of this scoping review and how it has been used to sharpen the full review method. 

There are several grammatical issues that could be improved, but I expect a full revision is needed, so I do not provide line by line editorial remarks on this paper. Some key points are that the introduction is unnecessarily long. It is sufficient to say that the definition of disability is contested, and that understanding more about the promise of technology to promote work among persons with disabilities is timely and important. A third theme in the background is the breadth and complexity of the field, and the importance of integrative reviews like the one proposed (or the one completed, depending on your decisions). 

The volume and originality of the work is commendable, and the potential impact is high. I would strongly encourage the authors to take editorial direction to settle on a paper that suits "Disabilities". 

Reviewer 3 Report

This is a somewhat unusual paper in that it reports the results of a pilot study designed to inform a scoping study. As a demonstration of how to set the requirements and specifications of a scoping study, I think it does make a contribution to the literature and there is potential for publication.   However, I would recommend some changes:   The focus of the paper does shift around between the scoping study and the pilot. I think it would benefit the reader if there was a more definite focus on the pilot throughout the paper. This is particularly noticeable in the abstract and the methodology, where it is not clear if you are reporting the findings of the pilot or the full review. This confusion is not helped by the use of the past tense ('was conducted') for the pilot and the future tense for the full review ('we intend'). By the time we get to the findings, I'm really quite unsure of what you are reporting. If you are reporting on the pilot study, then keep the focus on that, and when you've fully described that process, then move on to discuss the implications for the scoping review. If this is the pilot and the scoping review, then focus on the review, showing how the pilot study fed into it.   The introduction would benefit from introducing the importance of scoping studies in examining the issues you present. This would then give you a platform to describe the contribution of this paper.      Your discussion is largely directed toward the literature on the local findings (ie why classification is problematic; abstracts; and the future study). However, the point of the paper appears to be on demonstrating the role of piloting in scoping reviews, but this is only dealt with in a few lines - 255-260 - and is very descriptive. I think you need to discuss the implications of piloting with greater reference to the literature on scoping studies.   As I say, I think this paper could make a useful contribution to the literature, but it needs a clearer focus. Otherwise, it is a little confusing for the reader.  

Round 2

Reviewer 3 Report

I think the additions make the purpose clearer in terms of the scope of the paper.

I would, however, suggest making two very minor revisions. First, a small addition at 109 at the end of the 1.3 that clearly outlines what the paper will 'do' (reflecting the additions in the conclusion). It should only take a couple of sentences - 'this paper will outline the protocol for the review and demonstrate how piloting can be used to inform the development of the protocol'.  

Second, you might also wish to revisit the abstract. Currently, the emphasis is on the protocol rather than the paper itself. Again, it would only really need a sentence or two similar to the one above (I'd imagine in place of the third sentence).

Author Response

We would like to thank the Reviewer for their time and valuable feedback.

As suggested, we added the additional sentences to the abstract and at the end of the introduction. Please see lines 12-13 and 110-111. The changes are highlighted in yellow for your convenience.